# Dual Functional S-Doped g-C_3_N_4_ Pinhole Porous Nanosheets for Selective Fluorescence Sensing of Ag^+^ and Visible-Light Photocatalysis of Dyes

**DOI:** 10.3390/molecules24030450

**Published:** 2019-01-27

**Authors:** Abhijit N. Kadam, Md. Moniruzzaman, Sang-Wha Lee

**Affiliations:** Department of Chemical and Biological Engineering, Gachon University, Seongnamdaero 1342, Seongnam-si, Korea; abhikadamchem@gmail.com (A.N.K.); mani57chem@gmail.com (M.M.)

**Keywords:** S-doped g-C_3_N_4_, pinhole porous nanosheet, Ag^+^ ions, fluorescence sensing, visible light, photocatalytic degradation, cationic dyes

## Abstract

This study explores the facile, template-free synthesis of S-doped g-C_3_N_4_ pinhole nanosheets (SCNPNS) with porous structure for fluorescence sensing of Ag^+^ ions and visible-light photocatalysis of dyes. As-synthesized SCNPNS samples were characterized by various analytical tools such as XRD, FT-IR, TEM, BET, XPS, and UV–vis spectroscopy. At optimal conditions, the detection linear range for Ag^+^ was found to be from 0 to 1000 nM, showing the limit of detection (LOD) of 57 nM. The SCNPNS exhibited highly sensitive and selective detection of Ag^+^ due to a significant fluorescence quenching via photo-induced electron transfer through Ag^+^–SCNPNS complex. Moreover, the SCNPNS exhibited 90% degradation for cationic methylene blue (MB) dye within 180 min under visible light. The enhanced photocatalytic activity of the SCNPNS was attributed to its negative zeta potential for electrostatic interaction with cationic dyes, and the pinhole porous structure can provide more active sites which can induce faster transport of the charge carrier over the surface. Our SCNPNS is proposed as an environmental safety tool due to several advantages, such as low cost, facile preparation, selective recognition of Ag^+^ ions, and efficient photocatalytic degradation of cationic dyes under visible light.

## 1. Introduction

Environmental pollution has received substantial attention due to rapid industrialization; toxic pollutants such as heavy metals and dyes that are continuously released, posing severe risks to living organisms commonly existing in the environment [1,2,3]. Therefore, it is urgent to explore simple, reliable, and efficient techniques which can detect heavy metal ions and removing dyes from aqueous solutions to defend water assets and food provisions [4,5,6].

Silver is a widely used material owing to its high thermal and electric conductivity [7]. Almost 2500 tons of silver have been released into the environment every year because of its wide use in a number of industries, such as agriculture, pharmaceuticals, mirrors, and foods [8]. It was discovered that Ag^+^ can bind through several metabolites and enzymes, consequently leading to the obstruction of their normal functions [9]. Nowadays, silver is considered one of the most harmful and worldwide contaminants, with a simple impact on the aquatic biota and public health [10]. Therefore, the selective recognition of Ag^+^ offers significant advantages so that many different analytical approaches have been investigated [11,12]. However, most of these methods have common limitations such as high cost, time consumption, and low sensitivity [13]. Appropriate fluorescent probes which can address these practical concerns are in high demand [14,15,16]. The majority of fluorescent probes are employed as metal nanoparticles and metal chalcogenides, needing further surface modification to mitigate their own toxicity reflections. Thus, facile, efficient, and robust detection of Ag^+^ ions are still challenging tasks. It is extremely important to fabricate simple, low-cost, metal-free sensors to promote environmental and human well-being.

Recently, graphitic carbon nitride (g-C_3_N_4_) as a metal-free material has attracted significant interest in various research fields, such as florescence probing, photocatalysis, and gas sensing, because the g-C_3_N_4_ possesses an appropriately low band gap, chemical and thermal stability, high fluorescence, and excellent biocompatibility, as well as being prepared using readily available raw materials [17,18,19,20]. This material can also be used as an efficient photocatalyst for dye degradation. Apart from heavy metal ions, dyes are also a serious pollutant in aquatic environments due to their persistent color, toxic nature even in dilute concentration, and potential carcinogenicity [21]. Thus, photocatalysis has gained much attention due to its unique advantage, e.g., its ability to degrade the dye pollutant with low concertation into less toxic molecules (such as CO_2_, minerals, and H_2_O) under an ambient sunlight condition [22]. In this regard, dual-function materials with sensing capability and visible-light photocatalysis would play a crucial role in environmental monitoring and photocatalytic degradation of hazardous chemicals which are a big threat to environmental safety [23].

However, conventional g-C_3_N_4_ undergoes rapid recombination and insufficient utilization of visible light. Hence, the g-C_3_N_4_ doped with foreign atoms is of significant interest because doping process can tune the chemical nature of the g-C_3_N_4_, i.e., reducing its defect densities and modifying its textural, chemical, and optical properties [24]. Doping with non-metal impurities narrows the band gap of the g-C_3_N_4_ and stimulates its photocatalytic performance [25]. In particularly, S atoms can substitute for the N atoms of g-C_3_N_4_ due to suitable electronegativity and ionic radius, consequently leading to more favorable redox properties for enhanced photocatalytic activity [26]. Liu et al. [27] synthesized S-doped g-C_3_N_4_ from as-synthesized g-C_3_N_4_ powder calcined at 450 °C under a high-purity H_2_S gaseous atmosphere. But, this method releases a significant amount of poisonous gas during synthesis, which may cause a serious hazard in the surrounding environment. Jourshabani et al. [28] synthesized S-doped g-C_3_N_4_ to improve photocatalytic activity by using both thiourea and SiO_2_ as a sulfur source and a template, respectively. On the other hand, Ge et al. [29] prepared S-doped g-C_3_N_4_ with enhanced photocatalytic activity by using thiourea as a sulfur source without a template. The thiourea (or urea) is an inexpensive and easily available precursor for the in situ synthesis of S-doped g-C_3_N_4_ semiconductors.

In this work, the S-doped g-C_3_N_4_ (SCNPNS) with porous structure was successfully prepared by a facile heating method using inexpensive and easily accessible urea and thiourea precursors. Thiourea was used as a source for sulfur. The as-synthesized SCNPNS served as an effective fluorescent probe for selective recognition of Ag^+^ and visible-light active photocatalyst. The photocatalytic performance was tested through the degradation of methylene blue (MB) as a cationic dye and methyl orange (MO) as an anionic dye under visible light. A plausible sensing and photocatalytic mechanism were also explored in the present work.

## 2. Results and Discussion

### 2.1. Structure Characterization of SCNPNS

XRD analysis of g-C_3_N_4_ was performed to determine the effect of sulfur-doping on the phase purity and crystal structure. Figure 1a shows the XRD patterns of pure g-C_3_N_4_ and S-doped g-C_3_N_4_ (SCNPNS). The pure g-C_3_N_4_ exhibited two characteristic peaks at about 13.08° and 27.8°. These two peaks are in good accordance with those of g-C_3_N_4_ [30,31]. The distinct peak at 27.8° is indexed to the (002) plane that was attributed to the stacking of the conjugated aromatic system [31,32]. The minor peak at 13.08° is indexed to the (100) plane of g-C_3_N_4_, which is derived from repeated in-planar tri-s-triazine units [32,33]. The main peak of the SCNPNS was slightly shifted to a lower angle, which could be ascribed to the substitution of sulfur in the lattice atoms of carbon nitride [29]. Moreover, as compared to that of the pure g-C_3_N_4_, the peak intensity of SCNPNS was slightly decreased after sulfur doping, suggesting the exfoliation of bulk g-C_3_N_4_ [30,31,32,33].

FT-IR analysis was conducted to study the surface functional groups and nature of chemical bonding in SCNPNS, as shown in Figure 1b. The FT-IR spectrum of SCNPNS displays a broad band in the range of 3000 to 3500 cm^−1^, which was attributed to the NH and OH vibrational stretching modes, confirming the presence of –NH, –NH_2_, and hydroxyl groups of the adsorbed H_2_O [34]. The distinct peaks in the range of 1200 to 1700 cm^−1^ (including 1250, 1324, 1411, 1454, 1571, and 1624 cm^−1^) were corresponding to the typical stretching vibration modes of CN heterocycles [35]. The peak at 804 cm^−1^ belongs to the breathing mode of the triazine units, which was associated with the condensed CN heterocycles [36]. The vibration band at 889 cm^−1^ was ascribed to the cross-linked heptazine deformation of NH [37]. The sulfur doping was revealed by the peak at 705 cm^−1^, which was attributed to the C–S stretching vibration [38].

Figure 1c,d show the TEM images of SCNPNS at different magnifications, and the magnified TEM image of Figure 1d clearly indicates the presence of pinholes in the nanosheet. A pinhole structure was clearly observed from the TEM image recorded at the 20-nm scale. The pinhole structure could be attributed to the release of gases during heating and condensation, leading to the formation of pinhole-like nanosheets. The appearance of the pinhole structure reveals the porous nature of SCNPNS, which is beneficial for sensing and photocatalysis applications because a porous structure offers more active sites.

The N_2_ adsorption–desorption isotherms and the pore size distribution of the SCNPNS are shown in Figure 2. The isotherm curve of the SCNPNS is classified as type IV adsorption isotherm with H3 hysteresis loop at high pressure, signifying the network of the porous structure (Figure 2a). Based on the desorption curve, the BET surface area of SCNPNS is calculated to be 60.2 m^2^ g^−1^. The Barret–Joyner–Halenda (BJH) pore size distribution of SCNPNS is illustrated in Figure 2b. The average pore size is ca. 39 nm with a pore volume of 0.59 cm^3^/g, suggesting the mesoporous nature of as-synthesized SCNPNS. The mesoporous structure can provide higher surface area and more active sites, which are very useful for environmental applications such as adsorption, photocatalysis, and gas sensing.

An XPS analysis was performed to reveal further details on the chemical composition and oxidation state of SCNPNS prepared at 520 °C. An XPS survey spectrum confirms the presence of carbon, nitrogen, and oxygen atoms (data not shown). The intensity of the N peak shows high intensity due to the maximal atomic ratio of N in the SCNPNS. In addition, there was no distinct peak of sulfur (S) in the survey spectrum of SCNPNS because of the very low contents of S in the sample (0.1%). Figure 3 shows the core spectra of C 1s, N 1s, O 1s, and S 2p_1/2_, in order to investigate the bonding states of each element. The C 1s core spectrum is deconvoluted into four peaks centered at 284.6, 285.8, 287.8, and 288.5 eV (Figure 3a). The peak at 284.6 eV is assigned to sp^2^ graphite C=C bonds, whereas the peak at 285.8 eV is attributed to C–N group. The main strong peak at 287.8 eV is attributed to sp^2^ carbon–nitrogen (N–C=N) bonding of the aromatic ring system (such as the s-triazine unit and pyridine-like structure), whereas the peak at 288.5 eV is assigned to tertiary nitrogen N–(C)_3_ groups, i.e., nitrogen trigonally bonded to three sp^2^ carbon atoms in the C–N network [39,40,41]. The asymmetrical N 1s spectrum in Figure 3b can be deconvoluted into two peaks with binding energies at 398.4 eV and 400.4 eV, which are ascribed to sp^2^-hybridized nitrogen (C=N–C) and amino functional groups with a hydrogen atom (C–N–H), respectively [42]. As illustrated in Figure 3c, the core spectrum of O 1s has two symmetrical peaks around 532 eV and 534 eV, which are assigned as the existence of N–C–O groups and surface adsorbed water, respectively [43]. Figure 3d shows the core spectrum of S 2p_1/2_ at 163.9 eV, indicative of the C–S bond in SCNPNS with substituted sulfur for lattice nitrogen [44].

In summary, the XRD patterns of pure g-C_3_N_4_ and S-doped g-C_3_N_4_ (SCNPNS) exhibit two characteristic peaks at 13.08° and 27.8° corresponding to repeated in-planar tri-s-triazine units and the stacking of the conjugated aromatic system, respectively. FT-IR spectrum of SCNPNS shows the triazine units at 804 cm^−1^ and the C–S stretching vibration at 705 cm^−1^. XPS analysis exhibits the core-level spectrum of S 2p^1/2^ at 163.9 eV, indicative of the presence of C–S bonds in the SCNPNS. Interestingly, TEM analysis indicates the presence of pinholes on the nanosheet, which reveals the porous nature of the SCNPNS. From the BET measurements, the surface area, pore diameter, and pore volume of the SCNPNS were estimated to be 60.2 m^2^/g, 39 nm, and 0.59 cm^3^/g, respectively. Thus, overall characterization techniques (XRD, FT-IR and TEM, FT-IR, N_2_ adsorption isotherms and XPS) used in this study successfully confirm the presence of S in the framework of g-C_3_N_4_ and porous pinhole structure of the SCNPNS.

### 2.2. UV–Vis and Fluorescence Properties of SCNPNS

The optical properties of the SCNPNS were examined by UV–vis absorption, excitation, and emission fluorescence spectroscopy, as shown in Figure 4a. The SCNPNS exhibited a broad absorption peak at 330 nm which was bathochromically shifted, as compared to 310 nm of pure g-C_3_N_4_ (Appendix A) [45]. This bathochromic shift is mainly due to sulfur doping which can change the optical and electrical properties [28]. The photoluminescence (PL) spectra of the SCNPNS exhibited a strong PL emission peak at 445 nm under the 350-nm excitation. Furthermore, the inset of Figure 4a displays the different colors of SCNPNS solution under visible light and UV irradiation at 365 nm, respectively. The SCNPNS solution was nearly transparent under visible light, but produced a strong blue fluorescence under 365-nm UV light irradiation.

The excitation-independent fluorescence emission is observed in Figure 4b. With the increase of excitation wavelength from 290 to 380 nm by 10-nm intervals, the normalized PL intensity was increased for the excitation wavelength up to 350 nm, but a further increase in the excitation wavelength (>350 nm) resulted in a decrease in the PL intensity without the shift in the emission wavelength. Thus, the maximum excitation and emission wavelengths of the SCNPNS were found to be 350 nm and 445 nm, respectively. These excitation-independent fluorescence phenomena suggested that the SCNPNS exhibited a single-emission fluorescence center with high crystallinity [46]. As shown in Figure 4c, the PL intensity of the SCNPNS was stable without apparent photo-bleaching even under UV-light illumination with a high-pressure mercury vapor lamp for 300 min.

To obtain the optimal conditions for quantitative analytical application, the pH effect on the fluorescence intensity of the SCNPNS was investigated by varying the pH in the range of 2 to 13. The fluorescence intensity of the SCNPNS at various pH values is displayed in Figure 4d, in which the fluorescence intensity of the SCNPNS was found to be pH-dependent. The fluorescent intensity gradually increased as the pH increased from 2 to 7, whereas the intensity decreased abruptly at strongly basic pH (pH 10 to pH 12), probably due to the deprotonation of nitrogen in the basic environment. This may affect the distribution of excitons, resulting in the generation of low PL intensity [47]. A maximum value of 32% for the quenching of PL intensity was observed at pH 13, suggesting that our florescent probe (SCNPNS) is quite stable over a very wide range of pH 2 to 13. The highest PL intensity was observed at pH 7, which is coincident with the physiological pH condition. Therefore, neutral pH 7 was used in all fluorescence sensing experiments of the SCNPNS for the selective detection of various heavy metal ions.

### 2.3. Selective Fluorescence Sensing of SCNPNS toward Ag^+^

Figure 5 shows the fluorescent features of SCNPNS for 15 different biologically and environmentally-relevant metal ions (including Co^2+^, In^3+^, Ca^2+^, Cd^2+^, Fe^3+^, Cu^2+^, Fe^2+^, Blank, Ag^+^, K^+^, La^3+^, Mn^2+^, Na^+^, Hg^2+^, Pb^2+^, and Zn^2+^) at pH 7. After the addition of these heavy metal ions (20 μM), the pictorial image of the SCNPNS solution was recorded under UV light (365 nm). An aqueous solution of SCNPNS with Ag^+^ displayed significant fluorescent quenching, but no prominent fluorescence quenching was observed for other metal ions (Figure 5a).

The relative PL responses of blank SCNPNS and the SCNPNS after quenching with metal ions are shown in Figure 5b. Among all the metal ions, only Ag^+^ ions induced the substantial decrease in the fluorescence intensity of the SCNPNS. This result clearly indicated that the SCNPNS has high selectivity for Ag^+^ ions and can be used for the sensitive detection of Ag^+^. The high selectivity of the SCNPNS toward Ag^+^ was due to the synergistic effect of the N and S in SCNPNS, i.e., N in the SCNPNS had a strong chelating affinity with Ag^+^ [45]. To confirm high selectivity towards Ag^+^ ions, more metal ions were tested and are shown in Appendix A, which also confirmed that the SCNPNS is highly selective for Ag^+^ ions. We also carried out the control experiment for Ag^+^ sensing for pure g-C_3_N_4_ nanosheets. It was observed that quenching percentage of pure C_3_N_4_ was less as compared with that of SCNPNS for a similar concentration of Ag^+^ ions (please see Appendix A). This indicates that substitution of S plays an important role for Ag+ ions sensing. The detailed mechanism is discussed in the following Section 2.4.

Due to the good selectivity of Ag^+^, we investigated the fluorescence quenching efficiency of SCNPNS as a fluorescent probe for quantitative recognition of Ag^+^ under optimal conditions. As the concentration of Ag^+^ increased from 0 to 20 μM, the fluorescence intensity of SCNPNS decreased gradually, as shown in Figure 5c. The fluorescence intensity does not fit the linear calibration plots over the whole concentration range of Ag^+^ (from 0 to 20 μM), as shown in Figure 5d. However, the inset of Figure 5d, displayed good linearity of (F_0_ − F)/F_0_ over the relatively wide range of Ag^+^ from 0 to 1 μM (1000 nM). This concentration linearity confirms the reliability of our SCNPNS fluorescent probe for the sensitive detection of Ag^+^. The linearly regressed equation was (F_0_ − F)/F_0_ = −0.02 + 0.259 × C with R^2^ = 0.9979, where F_0_ and F are the fluorescence intensities of the SCNPNS solution in the absence and presence of Ag^+^, and C is the concentration of Ag^+^. The limit of detection (LOD) was determined as 57 nM at a signal-to-noise ratio of 3 using Equation (1):(1)$$LOD=\frac{3\sigma }{s}$$where σ and s are the standard deviation of the blank SCNPNS samples (*n* = 6), and the slope of the calibration plot, respectively. Table 1 lists the compared LOD between the present method and previously reported methods of numerous fluorescent probes for Ag^+^ detection. The LOD of the present work is comparable or superior to the previous reports [48,49,50,51,52,53,54,55]. In reference, the sensitivity requirement of Ag^+^ recognition in drinking water is 460 nM (defined by U.S. Environmental Protection Agency), indicative of the validity of our SCNPNS probe in the practical application [56]. The main advantage of the present work is that the SCNPNS is a metal-free fluorescent probe and prepared using easily accessible sources of urea and thiourea.

### 2.4. Plausible Mechanism of PL Quenching of SCNPNS with Ag^+^ Ions

The quenching mechanism of the SCNPNS in the presence of Ag^+^ ions can be elucidated with the help of UV–visible absorption, zeta potential, and the study of lifetime decay. Figure 6a shows that the addition of Ag^+^ to SCNPNS induces the bathochromic absorption spectral shift because of the formation of SCNPN-Ag^+^ complex. The change in UV–visible absorption spectra confirms the strong binding affinity of Ag^+^ onto the surface of SCNPNS. These results can be further explained by the concept of hard and soft acids and bases (HSAB), i.e., Ag^+^ is a soft acid and nitrogen (pyridine N and graphitic N) and sulfur in SCNPNS are soft bases [48]. According to HSAB, soft acids have a strong interaction with soft bases. Therefore, Ag^+^ ions may have a strong affinity with SCNPNS. Finally, the PL quenching mechanism of SCNPNS was clarified by time-resolved fluorescence lifetime measurements. The lifetime fluorescence decay of all the curves can be examined by fitting in a double exponential function based on the following Equation (2).
(2)$$D\left(t\right)=\overset{n}{\underset{i=1}{\sum }}\mathrm{ai}exp\left(-cos\frac{t}{T_{i}}\right)$$where *D* is the decay of PL, *t* is lifetime, *T_i_* is the PL lifetimes of several forms of fluorescent, and *ai* is the respective pre-exponential factor. The fluorescence decay curves are shown in Figure 6b, revealing no change in the fluorescence lifetime value of SCNPNS without (3.89 ns) and with the addition of 20-μM Ag^+^ ions (3.83 ns). Thus, the spectral shift in the absorption spectrum and the lack of change in lifetime indicate the formation of a ground-state non-fluorescent complex between SCNPNS and Ag^+^ ions [57,58].

Appendix A shows the double exponential fitting parameters of fluorescence lifetime decay curves of SCNPNS before and after adding of Ag^+^ ions. Moreover, Ag^+^ has a lower redox potential than the conduction band (CB) of SCNPNS; as a result, the photoinduced electron transfer (PET) from the CB of SCNPNS to Ag^+^ led to fluorescence quenching [45,57,59]. To prove the photoinduced electron transfer, we recorded a pictorial image of all metal ions (including Ag^+^ ion) in g-C_3_N_4_ under UV light irradiation for 1 min. The selective color change for Ag^+^ ions from colorless to brown indicates that Ag^+^ ions adsorbed on the surface get reduced due to photoelectron transfer (Appendix A). The proposed mechanism of the SCNPNS interaction with the Ag^+^ ions is graphically illustrated in Scheme 1.

### 2.5. Photocatalytic Degradation of Dyes under Visible Light

To examine the photocatalytic performance of the SCNPNS under visible light, an anionic MO dye and a cationic MB dye were used as representative organic dye pollutants. In addition, blank experiments such as photolysis and adsorption were conducted to examine the accurate photocatalytic activity of the SCNPNS. Figure 7a shows the variation of C/C_0_ in MB and MO as a function of time under various conditions. The results include photolysis, adsorption in the dark on the SCNPNS, and the photocatalytic activity of SCNPNS under visible light illumination. The photolysis of both MO and MB dyes was negligible without SCNPNS photocatalysts. However, distinct differences of adsorption and photocatalytic degradation of MB and MO dyes over the SCNPNS are apparent. Adsorption of MO on the surface of SCNPNS was almost negligible, but 34% of MB was adsorbed onto the SCNPNS in dark condition. To explore the reason for the different adsorption performance of MB and MO dyes, the zeta potential of SCNPNS was measured using a zeta analyzer.

The zeta potential of the pristine SCNPNS was measured as −7.23 mV and increased to +15.31 mV by the addition of Ag^+^ ions, confirming the negative charges of the SCNPNS (Appendix A). Since MB is a cationic dye and MO is an anionic dye, different charge properties and porous structure may have affected the adsorptive properties, which play a vital role in the manifestation of superior photocatalytic properties.

According to Figure 7a, the photocatalytic performance toward MB was significantly superior to the degradation of MO. After 180 min of photocatalytic reaction, 90% of MB was degraded, whereas only 14% of MO was degraded under identical conditions. Figure 7b,c display the variation in the characteristic absorption of MB and MO over the SCNPNS under visible light, respectively. MB almost disappeared within 180 min, whereas the absorbance of MO does not change significantly during the same length of adsorption times. No peak shift in the maximum absorption wavelength was observed for either dye, indicating a lack of other identifiable fragments [60].

The quantitative determination of the photocatalytic performance towards MB and MO dyes was investigated using the Langmuir–Hinshelwood kinetics model. Figure 7d displays the plot of ln(C_0_/C) versus time (t) for the photocatalytic performance. The linear relationship between ln(C_0_/C) and time indicates that photocatalytic degradation of MB and MO follows a pseudo-first-order kinetics [61]. The fitted kinetic rate constants of SCNPNS were found to be 0.01153 min^−1^ and 6.5 × 10^−4^ min^−1^ for the photocatalytic degradation of MB and MO dyes, respectively. The maximum photocatalytic performance was observed toward MB dye, with a rate constant of 18-fold higher than that of MO. The superior photocatalytic performance toward MB was ascribed to the negative charge and porous structure of SCNPNS, leading to the greater adsorption and superior photocatalytic activity toward the cationic MB dye as compared to the anionic MO dye.

After photocatalysis experiments under visible light irradiation of dye excitation, the catalyst was separated from the reaction mixture by centrifugation and washed with distilled water and ethanol until a clear supernatant was obtained. The recovered catalyst was dried at 80 °C for 1 h, and we carried out an XRD analysis of the recovered samples before and after photocatalysis, in order to ensure the structural stability of the photocatalyst. According to XRD data, the crystalline phase and structure of the SCNPNS after the reaction were almost identical with those of the pristine sample (Appendix A). This XRD result gave a good support for the photostability of the SCNPNS as visible-light photocatalyst.

The photocatalytic mechanism is illustrated in Scheme 2. During the adsorption–desorption equilibrium process, large amounts of MB was adsorbed on the surface of SCNPNS due to electrostatic interactions between the cationic MB dye and the negatively charged catalyst, which was confirmed by the zeta potential of SCNPNS of −7.23 mV. Hence, adsorption process is considered a crucial step for photocatalysis, responsible for efficient dye degradation. Similar synergistic effects have been reported for other photocatalysts (WO_3_/g-C_3_N_4_, *N*-doped CeO_2_, and carbon nitride nanowires/nanofibers) [62,63]. Moreover, Liu et al. [27] reported that sulfur doping is effective for narrowing band gap by the interaction of S 3p states with N 2p. The narrow band gap leads to enhanced absorption of visible light and more generation of electron-hole pairs [64]. When visible light is subjected to the SCNPNS, the electrons in the valence band (+1.57 eV vs. NHE) migrate to the conduction band (−1.13 eV vs. NHE), consequently producing photo-generated electrons and holes [29]. Because of the pinhole porous nanosheets, more electrons and holes can transfer to the surface of SCNPNS. The photo-generated electrons can react with dissolved oxygen to produce superoxide radical anions [62]. The super oxide radical anions then react with water to produce hydroxyl radicals. Finally, generated active radicals (such as super oxide anions and hydroxyl radicals) have strong oxidation potential enough to degrade dye molecules that are photosensitized under visible light. Because of the redox potential of OH/OH^−^ (2.7 V vs. NHE), the photo-generated holes on VB (1.57 eV) cannot directly oxidize water to produce OH radicals [65]. However, h^+^ can oxidize MB directly due to the comparatively small oxidizing potential of MB (1.25 V vs. NHE) [66].

## 3. Materials and Methods

### 3.1. Materials

Urea, thiourea, NaOH, HCl, CoCl_2_·6H_2_O, In(NO_3_)_3_·6H_2_O, Ca(NO_3_)_2_·4H_2_O, Cd(NO_3_)_2_·4H_2_O, FeCl_3_·6H_2_O, CuCl_2_·2H_2_O, FeCl_2_·4H_2_O, AgNO_3_, KCl, La(NO_3_)_3_·6H_2_O, MnCl_2_·4H_2_O, NaCl, Hg(NO_3_)_2_·H_2_O, Pb(NO_3_)_2_, ZnCl_2_·2H_2_O, NiCl_2_·6H_2_O, SnCl_4_·5H_2_O, ZrOCl_2_·8H_2_O, TiCl_3_, and Bi(NO_3_)_3_·5H_2_O were obtained from Sigma Aldrich, USA. All the reagents used in this study were of analytical grade and used without further purification. All the solutions were prepared in deionized water throughout this study.

### 3.2. Synthesis of the SCNPNS

Metal-free SCNPNS with porous structure were synthesized by following the previously reported facile heating method with slight modification [33]. In detail, 3 g of urea and 10 g of thiourea were ground in an agate mortar for 15 min. Afterwards, the powder was transferred into an alumina crucible, covered with a lid and alumina foil. Then, the covered alumina crucible was heated to 520 °C for 2 h at a ramping rate of 5 °C·min^−1^ in air atmosphere. Next, the crucible was cooled to room temperature (RT). The obtained product was collected and grounded into powder. In the next process, the 50-mg powder was dispersed in 50 mL of water by ultra-sonication for 2 h. Subsequently, the suspension was centrifuged at 6000 rpm for 10 min, and the supernatant liquid as a stable solution was further used as a stock solution for detection of Ag^+^ ions in water.

### 3.3. Instrumental Characterization

In order to investigate crystal structure and phase purity of the SCNPNS, the powder X-ray diffraction (XRD) patterns were recorded on (Rigaku, Cu-Kα radiation, D/max-2200pc, Rigaku, Tokyo, Japan, λ = 1.5418 Å) with an accelerating operating voltage and applied current was of 40 kV and 30 mA, respectively. The surface morphology of SCNPNS was analyzed by transmission electron microscopy (Tecnai, G2mF30 S-Twin AP Tech., Hillsboro, OR, USA, 200 kV). The functional groups were investigated by Fourier transform infrared (FT-IR) spectroscopy (Bruker, Vertex 70 FT-IR spectrometer, 104-0033 Tokyo, Japan). N_2_ adsorption–desorption was measured using the Brunauer–Emmett–Teller (BET) method by a Micromeritics ASAP-2010 surface area analyzer (Corp., Norcross, GA, USA). To examine the surface composition and oxidation state of SCNPNS, the XPS analysis was carried out using ESCALAB220i (VG Scientific, Thermo. Scientific, East Grinstead, UK) on the sample with an Al K-alpha radiation X-ray source. The pH of solutions was recorded by using a digital pH meter (OHAUS-STARTER1100 Instrument Co., Ltd., Shanghai, China). Ultraviolet-visible (UV–vis) absorption spectra were noted using Varian Cary 100 UV–vis spectrophotometer (Markham, ON, Canada). Fluorescence emission spectra were recorded at RT with an excitation wavelength of 350 nm by using a fluoroluminescence spectrometer (Quanta Master, Photon Technology International, Lawrenceville, New Jersey, USA), equipped with an Xe lamp (Arc Lamp Housing, A-1010B™, NJ, USA,), monochromator, and power supply (Brytexbox, NJ, USA). Fluorescence lifetime decay curves were obtained by Easy Life II fluorometer system (Horiba, Ltd., Kyoto, Japan). The instrument response function (IRF) was obtained using a 1 wt% LUDOX^®^ LS colloidal silica suspension in water as the reference. An ELSZ-2000 (Otsuka Electronics Co., Ltd. Osaka, Japan) series zeta-potential and particle-size analyzer as acquired to examine the surface charge distribution of the SCNPNS.

### 3.4. Fluorescence Detection of Ag^+^ Ions

The selectivity of SCNPNS towards environmentally and biologically different metal ions (Co^2+^, In^3+^, Ca^2+^, Cd^2+^, Fe^3+^, Cu^2+^, Fe^2+^, Blank, Ag^+^, K^+^, La^3+^, Mn^2+^, Na^+^, Hg^2+^, Pb^2+^, Zn^2+^, Ni^2+^, Bi^2+^, Sn^4+^, Ti^3+^, and Zr^4+^) were investigated at room temperature (RT) under identical conditions. During a typical experiment, 100 μL of the SCNPN solution was added to 3 mL of 20 μM metal-ion solution. The reaction mixture was incubated for 5 min at RT, and fluorescent spectra were recorded under excitation at λ = 350nm. To study the Ag^+^ sensing capacity of SCNPNS, different concentrations of Ag^+^ (0 to 20 μM) were analyzed in the above manner. The fluorescence detection was conducted at RT and pH 7. A reference solution was also prepared without adding any ions.

### 3.5. Photocatalytic Activity of the SCNPNS

The photocatalytic activity of SCNPNS with respect to methylene blue as a cationic dye and methyl orange as an anionic dye was tested under visible light. To activate the photocatalytic reaction, a 100-W halogen lamp equipped with a UV filter by UV cut-off filter (JB-420, Nantong Haisheng Optical Co., Lt, Shanghai, China) was used as a visible light irradiation source. The photoreactor was placed on the magnetic stirrer in open air to gain enough oxygen for the photodegradation reaction. The temperature of the reactor was maintained constant by water circulation in an outer jacket surrounding the photoreactor. Typically, 50 mg of photocatalyst was dispersed in 100 mL of 10-ppm dye solutions. Before visible-light irradiation, the mixture was continuously stirred for 30 min in the dark to establish an adsorption–desorption equilibrium, and then the light source was triggered on. At a predetermined time interval, 3 mL of the sample solution was collected during the experiment and subsequently centrifuged to remove the photocatalyst. Finally, the sample solution was analyzed to estimate the extents of photodegradation through the absorbance intensity of testing dye (MB or MO) at a maximum absorption wavelength by UV–vis spectrophotometer.

## 4. Conclusions

Metal-free S-doped g-C_3_N_4_ pinhole nanosheets (SCNPNS) with porous structure were successfully prepared by a simple heating method using easily accessible precursors such as urea and thiourea. Various analytical tools (including XRD, XPS, FT-IR analysis) confirmed the successful doping of sulfur in g-C_3_N_4_ (i.e., S atoms replaced by lattice N atoms), forming C–S bond in g-C_3_N_4_. The pinhole porous structure of SCNPNS was clearly revealed by TEM analysis. This work demonstrates that the SCNPNS is a very promising candidate for application in selective and sensitive recognition of Ag^+^, with a detection limit of 0.57 nM, which covers the standard concentration range of silver ions in real samples. The fluorescence-quenching process described here is purely static, according to the UV–visible absorption shift, zeta potential, and lack of change in lifetime. Moreover, the photocatalytic activity of SCNPNS towards cationic and anionic dyes (MB and MO) was explored under visible light irradiation. The highest photocatalytic performance was observed for cationic MB dye, in which the rate constant was ~18 times higher than that of MO. It was found that the synergy between adsorption and photocatalysis (supported by zeta potential) and porous pinhole structure were responsible for efficient photocatalytic degradation of cationic MB dye. Conclusively, S-doped g-C_3_N_4_ (SCNPNS) showed numerous advantages, such as inexpensive, easy preparation, sensitive and selective detection of Ag^+^ ions, and efficient degradation of cationic dye. These features allow SCNPNS as a dual functional candidate for sensing and photocatalytic applications for environmental safety.

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
