# Peer review of "Dual Functional S-Doped g-C3N4 Pinhole Porous Nanosheets for Selective Fluorescence Sensing of Ag+ and Visible-Light Photocatalysis of Dyes"

_molecules, 2019, doi:10.3390/molecules24030450_

Round 1
Reviewer 1 Report
This manuscript by Lee et al. described the synthesis of S doped g-C3N4 material and tested for detection of Ag+ ions and photocatalytic decomposition of organic dye. Overall, the manuscript is well written and the claims are well supported by the experimental results. This paper is suitable to publish in Molecules as an article. I have some minor comments-
1) Did the authors have done any control study on unsubstituted C3N4 material for Ag sensing?
2) How about the residual emission after Ag ion addition, do the authors have any explanation?
3) What is the Stern-Volmer quenching constant for the emission quenching?
4) Do the authors have done any control study on the mechanism of dye degradation mechanism, such as- a) to understand the effect of oxygen, doing the reaction in absence of oxygen,
b) Reaction in absence of light?
c) PXRD after the catalysis?
5) Do they measure any surface area of the S doped g- C3N4 material?
6) Figure 3c- To measure the stability of the material under light excitation, the y axis should be emission maxima intensity, not the normalized intensity. If they normalized with respect to the maximum, then it doesn’t reflect the stability.
Author Response
Please, refer to the attached file.

Reviewer 2 Report
Dual functional S-doped g-C3N4 pinhole porous nanosheets for selective fluorescence sensing of Ag+ and visible-light photocatalysis of dyes
N. Kadam, Md. Moniruzzaman, Sang-Wha Lee*
Department of Chemical and Biological Engineering, Gachon University, Korea
The paper describes the development of an S-doped “pinhole-containing” g-C3N4 nanosheet material, which apparently can be obtained in larger quantities and as a stable aqueous solution. S is introduced as thiourea in the synthesis. The synthesis protocol is very simple: heating of thiourea or urea forms the graphitic carbonitride on heating to 520 oC. The S-doped g-C3N4 is a photo-active material. A number of methods are used to characterize the material: XPS (C; N; O; S); XRD, IR, electron microscope; UV-vis and fluorescence; ζ-potential. Photocatalytic dye degradation under visible light was tested with methylene blue (cationic) and methylorange (anionic).
Fig. 1 (top left): if one assumes that the intensity scale is not changed between the two spectra then the (002) interlayer signal is attenuated by about 50%. Does this represent a significant decrease? It is difficult to normalize the signal because the peak associated with the (001) plane is very weak..
The O signal in XPS is rather pronounced, even to there should be no O in the structure. In contrast, the sulfur XPS signal is weak, but a substantial amount of sulfur is introduced with the precursor. The authors should explain how much S is incorporated into the structure, and at what positions the sulfur substitution takes place. A bathochromic shift due to sulfur doping was observed, and it is postulated (without further proof) that sulfur doping changes the optical and electrical properties. Fluorescence spectra were measured as a function of pH.
However, the characterization of the pores is rather wanting: the authors provide only one electron micrograph, and it is rather difficult to see the pinholes to agree with the authors. In the magnified circle, is the darker contrast to be interpreted as holes and the uniform grey as nanosheet? The yellow circle on the less magnified frame has a diameter of 400 nm, whereas the yellow circle on the right photo, which should represent the same area, has a diameter of 100 nm.
What is the concentration of nanosheets in the aqueous solution? The authors start from 13 g of urea/thiourea, and take the resulting solid (mass not specified) up in water. The supernatant after centrifuging of the insoluble solids is the nanosheet solution. However, no information is given on the concentration. The high selectivity for silver is surprising, and the authors should check a few other ions like Tl+. Ions that bind strongly to S (and form very insoluble sulfides) such as Hg2+ and Pb2= surprisingly show little quenching.
Selective fluorescence quenching: Ag+ quenches, all other materials not. Formation of a SCNPN-Ag+ complex; spectral changes indicate the strong binding affinity of Ag to the SCNPN. Concept of hard and soft acids and bases (HSAB) – Ag+ is soft acid (electron acceptor) and N and particularly S are soft bases.
Absorption spectrum changes on adsorption of Ag, but the fluorescence lifetime remains unchanged. I do not know how the life time is defined, but the half life looks like 7 ns, rather than 3.83 ns (except if the IRF is subtracted as instrumental broadening (IRF = instrument response function, explained in line 334).
In 3.1. Materials, the yield of soluble g-C3N4 (starting from 13 g urea/thiourea) should be given.
The English is very poor, particularly in the early part of the manuscript; the reviewer has highlighted only a selection of poor formulations and wrong choice of vocabulary.
21 endorsed è proposed
29 heavy metal ions and dyes continuously release toxic pollutants, è due to rapid industrialization, toxic pollutants such as heavy metals and dyes are continuously released,
33 Silver has fascinated significant interest è Silver is a widely used material owing….
39 Therefore, selective recognition of Ag+ is an extremely imperative task so that many endeavors have been made è Therefore, the selective recognition of Ag+ offers significant advantages and many different analytical approaches have been investigated.
41 As a result, appropriate fluorescent probes are highly demanding which can address these practical concerns è Appropriate fluorescent probes which can address these practical concerns are in high demand.
49 represents è possesses
52 rigorous è persistent
55 minerals è the term “mineralization” effectively means to decompose to CO2 and water; unless the dye is a salt, there are no minerals in the molecule.
312 were grounded in mortar pestle for 15 min è were ground in an agate mortar; was shifted è was transferred
The symbol μ did not reproduce (175, 195, 197; 227)
Author Response
Please, refer to the attached file.

Reviewer 3 Report
The manuscript deals with the synthesis and characterization of S-doped g-C3N4 pinhole based material (SCNPNS) and its use as fluorescence sensor for Ag+ and as visible-light photocatalyst for dyes. The here prepared SCNPNS sample has been analysed by different techniques (i.e. XRD, XPS, UV-Visible, TEM, etc.) to demonstrate the incorporation of S-groups and the structure of the solid material. Fluorescence sensor capacity for Ag+ and photocatalytic activity for MB degradation have been studied and demonstrated. In general, the paper is well written, its quality and originality is adequate, and the topic is interesting from both scientific and technological viewpoints. Nevertheless, some unclear concepts must be addressed by the authors, while additional information (mainly related to SCNPNS material characterization) is needed to clarify and explain some affirmations made in this study, as well as to emphasize the significance of the results (see major remarks listed below).
Major comments and remarks:
i) The title states that the SCNPNS material possesses some porosity, but the authors give any data endorsing this fact. Nitrogen (or Argon) adsorption isotherms measurements could provide not only the values of specific surface area of the material, but also the type of porosity and the pore volumes calculated values (trough the BET method). Authors must address this important point.
ii) Elemental Analysis (EA) data could be also provided by the authors to ascertain the amount of S species incorporated onto the solid material, as well as the obtained chemical composition in terms of C, N and S molar ratios.
iii) Maybe a final paragraph (some few lines) summarizing the main structural and physicochemical properties measured and elucidated in section 2.1. Structure characterization of SCNPNS should be written by the authors for a better reader`s understanding.
iv) Some affirmations made by the authors should be revised without the information asked in the previous remarks (i) and (ii), this meaning if the authors are not providing the adequate data. Some examples are:
- “Metal-free S-doped g-C3N4 pinhole nanosheets (SCNPNS) with porous structure…”
- “The pinhole porous structure of SCNPNS was clearly revealed by TEM analysis.”
v) Other minor issues:
- In the last paragraph of section 1. Introduction, authors say that: “… urea and thiourea as sulfur precursors…” In fact, urea (just possessing C, N and H elements) could not be considered as a sulfur precursor. Authors must solve this mistake.
- Other type-setting faults appear all along the text. Authors must revise them.
Summarizing, I could consider the manuscript suitable for publication in this journal but only after the authors have made the changes and corrections following the remarks afore-mentioned.
Author Response
Please, refer to the attached file.

Round 2
Reviewer 2 Report
The paper has been revised to the satisfaction of the reviewer.
Please correct the following in the indicated lines:
75 ….using {deleted} inexpensive
76 {delete} Thiourea was used…..
89 … was shifted to a lower angle {delete}, which…..
116 the specific surface area should be given with no more than 3 significant digits (i.e., 60.2 m2/g)
141 can be deconvoluted into two {delete} peaks
Author Response
Dear Editor,
Thanks for your warm consideration for our manuscript entitled “Dual functional S-doped g-C3N4 pinhole porous nanosheets for selective fluorescence sensing of Ag+ and visible-light photocatalysis of dyes". We have revised the manuscript in the light of useful suggestions of the reviewers. All the necessary changes are made in the revised manuscript and changes are highlighted. More detailed responses to the specific comments and major changes are described as follows:
Reviewer#2: Comments and Suggestions for Authors
The paper has been revised to the satisfaction of the reviewer.
Please correct the following in the indicated lines:
75 ….using {deleted} inexpensive Þ Corrected
76 {delete} Thiourea was used….. Þ Corrected
89 … was shifted to a lower angle {delete}, which….. Þ Corrected
116 the specific surface area should be given with no more than 3 significant digits (i.e., 60.2 m2/g) Þ Corrected
141 can be deconvoluted into two {delete} peaks Þ Corrected
Reviewer 3 Report
After revision, authors have answered most of the questions and remarks made by this reviewer, but one issue remains unsolved. Thus, points (i), (ii), (iv), and (v) have been adequately answered by the authors, while remark (iii) remains unanswered, and further inputs are needed. Each one of the authors’ responses are commented below:
Reviewer comment to authors’ answer to point (i): Ok.
Reviewer comment to authors’ answer to point (ii): Ok.
Reviewer comment to authors’ answer to point (iii): Not answered.
Authors have slightly modified the already existing paragraph where they discusse the XPS results, but they have not added a new and last paragraph summarizing the main structural (XRD, FT-IR and TEM) and physicochemical (FT-IR, N2 adsorption isotherms and XPS) properties of SCNPNS material measured and elucidated in section 2.1. Authors should include the suggested paragraph.
Reviewer comment to authors’ answer to point (iv): Ok.
Reviewer comment to authors’ answer to point (v): Ok.
Summarizing, I consider the manuscript suitable for publication in this journal, only minor corrections by the authors are needed.
Author Response
Dear Editor,
Thanks for your warm consideration for our manuscript. We have revised the manuscript in the light of useful suggestions of the reviewers. All the necessary changes are made in the revised manuscript and changes are highlighted. More detailed responses to the specific comments and major changes are described as follows:
Reviewer#3: Comments and Suggestions for Authors
After revision, authors have answered most of the questions and remarks made by this reviewer, but one issue remains unsolved. Thus, points (i), (ii), (iv), and (v) have been adequately answered by the authors, while remark (iii) remains unanswered, and further inputs are needed. Each one of the authors’ responses are commented below:
Reviewer comment to authors’ answer to point (i): Ok.
Reviewer comment to authors’ answer to point (ii): Ok.
Reviewer comment to authors’ answer to point (iii): Not answered.
Authors have slightly modified the already existing paragraph where they discussed the XPS results, but they have not added a new and last paragraph summarizing the main structural (XRD, FT-IR and TEM) and physicochemical (FT-IR, N2adsorption isotherms and XPS) properties of SCNPNS material measured and elucidated in section 2.1. Authors should include the suggested paragraph.
Author response: In summary, the XRD patterns of pure g-C3N4 and S-doped g-C3N4 (SCNPNS) exhibited two characteristic peaks at 13.08° and 27.8° corresponding to repeated in-planar tri-s-triazine units and the stacking of the conjugated aromatic system, respectively. FT-IR spectrum of SCNPNS reveals the triazine units at 804 cm−1 and the C–S stretching vibration at 705 cm−1. XPS analysis exhibits the core-level spectrum of S 2p1/2 at 163.9 eV, indicative of the presence of C–S bonds in the SCNPNS. Interestingly, TEM analysis indicates the presence of pinholes on the nanosheet, which reveals the porous nature of the SCNPNS. From the BET measurements, the surface area, pore diameter, and pore volume were estimated to be 60.2 m2/g, 39 nm, and 0.59 cm3/g. Thus, overall characterization techniques (XRD, FT-IR and TEM, FT-IR, N2adsorption isotherms and XPS) used in this study successfully confirmed the presence of S in the framework of g-C3N4 and porous pinhole nanostructure of the SCNPNS.
Reviewer comment to authors’ answer to point (iv): Ok.
Reviewer comment to authors’ answer to point (v): Ok.
Summarizing, I consider the manuscript suitable for publication in this journal, only minor corrections by the authors are needed
Author response: Thank you very much for warm consideration of our manuscript. According to the
reviewer’s comments, we have added last paragraph, summarizing the main
structural (XRD, FT-IR and TEM) and physicochemical (FT-IR, N2adsorption isotherms and XPS) properties of SCNPNS materials in section 2.1.